# Whole-Genome Sequencing Can Identify Clinically Relevant Variants from a Single Sub-Punch of a Dried Blood Spot Specimen

**DOI:** 10.3390/ijns9030052

**Published:** 2023-09-21

**Authors:** David J. McBride, Claire Fielding, Taksina Newington, Alexandra Vatsiou, Harry Fischl, Maya Bajracharya, Vicki S. Thomson, Louise J. Fraser, Pauline A. Fujita, Jennifer Becq, Zoya Kingsbury, Mark T. Ross, Stuart J. Moat, Sian Morgan

**Affiliations:** 1Illumina Cambridge Ltd., Cambridge CB21 6DF, UK; 2Illumina Inc., San Diego, CA 92122, USA; 3Wales Newborn Screening Laboratory, University Hospital of Wales, Cardiff CF14 4XW, UK; 4School of Medicine, Cardiff University, Cardiff CF14 4XW, UK; 5All Wales Genetics Laboratory, University Hospital of Wales, Cardiff CF14 4XW, UK

**Keywords:** whole genome sequencing, newborn screening, dried blood spots, DNA sequencing, cystic fibrosis, phenylketonuria

## Abstract

The collection of dried blood spots (DBS) facilitates newborn screening for a variety of rare, but very serious conditions in healthcare systems around the world. Sub-punches of varying sizes (1.5–6 mm) can be taken from DBS specimens to use as inputs for a range of biochemical assays. Advances in DNA sequencing workflows allow whole-genome sequencing (WGS) libraries to be generated directly from inputs such as peripheral blood, saliva, and DBS. We compared WGS metrics obtained from libraries generated directly from DBS to those generated from DNA extracted from peripheral blood, the standard input for this type of assay. We explored the flexibility of DBS as an input for WGS by altering the punch number and size as inputs to the assay. We showed that WGS libraries can be successfully generated from a variety of DBS inputs, including a single 3 mm or 6 mm diameter punch, with equivalent data quality observed across a number of key metrics of importance in the detection of gene variants. We observed no difference in the performance of DBS and peripheral-blood-extracted DNA in the detection of likely pathogenic gene variants in samples taken from individuals with cystic fibrosis or phenylketonuria. WGS can be performed directly from DBS and is a powerful method for the rapid discovery of clinically relevant, disease-causing gene variants.

## 1. Introduction

Newborn screening (NBS) plays a vital role in healthcare systems for the prompt identification of individuals who may develop one of a set of rare, but severe health conditions [1]. In the UK, for example, dried blood spot (DBS) specimens are routinely collected at 4–5 days of age and are sent to regional screening laboratories for analysis. Sub-punches are taken from the DBS and analyzed for nine conditions “https://www.gov.uk/guidance/newborn-blood-spot-screening-programme-overview” (accessed on 1 September 2023).

Next-generation sequencing (NGS) has transformed genetic testing in the healthcare setting. Over the past 10 years, the technology has enabled clinical laboratories to progress from assays that focus on a single specific variant to assays that simultaneously query all positions in multiple genes or even whole genomes, dramatically changing what information a clinician can expect from a genetic test [2,3,4]. WGS is the most comprehensive test available to capture all classes of variants, from small events such as single-nucleotide variants (SNVs) and small indels, to larger changes in genetic material such as copy number changes (CNV) [5,6] and structural variants (SVs) [7]; the last two often originate deep in intronic regions not included on gene panels or whole-exome sequencing (WES). Calling SVs, even from WGS data, can be compromised by the presence of a repetitive or simple sequence involved in the variant, but there has been success in addressing such cases using combinations of callers or custom callers targeted at specific genomic regions [8,9,10]. The algorithms used to capture variants and distill the data into a clinical format are under continual improvement, with updates to analysis pipelines requiring validation as well as the laboratory workflows. While this rapid expansion of information can seem daunting, the content is incredibly powerful for aiding diagnosis. Indeed, in rare genetic disorders, whole-genome sequencing (WGS) has been established in numerous healthcare systems as a primary test, in a bid to lessen the diagnostic odyssey faced by patients with conditions which are challenging to diagnose [4,11,12,13,14,15,16]. While the value of such tests (and rapid diagnoses) for those who suffer from these conditions and their families is obvious “https://everylifefoundation.org/burden-study/” (accessed on 30 August 2023), “https://globalgenes.org/wp-content/uploads/2013/04/ShireReport-1.pdf” (accessed on 1 September 2023) [17,18,19,20,21,22], the impact of applying them more broadly across the population, or even how this might be accomplished, is less clear. The impact and feasibility of applying WGS to newborns are already being explored [12,21,23,24,25,26,27,28] “http://www.genomicsengland.co.uk/initiatives/newborns” (accessed on 2 August 2023, “https://www.iconseq.org/” (accessed on 1 August 2023), “https://babyscreen.mcri.edu.au/” (accessed on 31 August 2023).

One key consideration that is being explored in these studies is the source of DNA for WGS. Over the years, NGS workflows have evolved to accept smaller amounts of genetic material as input, but advancement has also been made in terms of allowing workflows to proceed directly from raw, unextracted samples such as blood (including DBS) or saliva. The removal of the extraction step from the workflow has the potential to reduce both the cost and the time taken to obtain the sequencing result while maximizing the DNA input to the assay from small amounts of material. Methods proceeding directly from DBS or DNA extracted from DBS have already been tested using both new and historical samples [25,29,30,31,32,33,34]. The number of sub-punches used by each group varies. In one study, the authors explored the DNA yields from different numbers of sub-punches, but did not compare WGS from these DNAs [29]. Limiting the number of DBS sub-punches is highly desirable: not only is there a considerable burden on the NBS pipeline in preparing repeated punches from the same sample, but different laboratories use different DBS punch configurations; therefore, the amount of material remaining after standard of care testing can vary. The generation of high-quality data from lower inputs would maximize the number of samples that could benefit from WGS testing.

In this study, we confirm that high-quality WGS data can be generated from libraries derived from DBS and investigate the feasibility of using fewer DBS sub-punches as inputs for detecting clinically relevant variants by WGS.

## 2. Materials and Methods

### 2.1. Samples

#### 2.1.1. Consent and Ethics Approval

The study was reviewed by the local research and development department and approved as a service evaluation; formal ethical approval was not required.

#### 2.1.2. Dried Blood Spot Collection

To compare WGS data generated from liquid whole and DBS specimens, we retrieved anonymized residual blood specimens collected into EDTA blood tubes. An aliquot of the liquid blood specimen was taken, and 50 mL aliquots were applied to filter paper collection devices (PerkinElmer 226, Perkin Elmer, Waltham, MA, USA). Liquid whole blood specimens from the PKU patients were collected during routine monitoring and genetic analysis as part of the KUVAN responsiveness testing pathway [35], and an additional sample was collected from these patients by spotting blood from a finger prick directly onto the blood collection card. The blood spot cards were stored at room temperature for 1 to 6 weeks before processing.

#### 2.1.3. DNA Extraction from Venous Blood

Nucleic acid extraction was undertaken from 2–4 mL of peripheral blood using the Chemagic STAR instrument (Perkin Elmer, Waltham, MA, USA), following the manufacturer’s protocol.

### 2.2. DNA Quantification

DNA samples extracted from blood were quantified on a Gemini XPS Microplate Reader (Molecular Devices, San Jose, CA, USA) using the Quant-iT™ PicoGreen^®^ dsDNA Assay Kit.

### 2.3. NGS Library Preparation

Sequencing libraries were prepared using the Illumina DNA PCR-Free Prep kit (Illumina Inc., San Diego, CA, USA), which proceeds directly from six 3.2 mm DBS sub-punches when following the standard protocol (Illumina DNA PCR-Free Library Prep Reference Guide, Illumina document #1000000086922 v03). The appropriate number of 3.2 mm or 6 mm sub-punches (6, 3 or 1) was obtained from each DBS using a DBS Puncher^®^ Instrument (Perkin Elmer, Waltham, MA, USA). DNA was released from the sub-punches by shaking at 1000 rpm at 56 °C on a ThermoMixer C (Eppendorf, Hamburg, Germany) in 300 µL of 1× Lysis Master Mix prepared from 10x Modified Lysis Buffer and proteinase K (2 µL), as described in the Illumina Lysis reagent kit protocol (Illumina, Inc., San Diego, CA, USA). After lysis was completed, 135 µL of Illumina Purification Beads were added, and the solution was vortexed briefly and left at room temperature for 5 min. Beads were collected by incubation on a magnetic stand for a further 5 min, then washed for a total of two 30 s washes in 500 µL of 80% EtOH. After the removal of the second wash, beads were air-dried for 5 min before elution in 35 µL of resuspension buffer (RSB). An optional sample QC step was performed by loading 1 µL of lysed material onto the Agilent Genomic DNA ScreenTape System (Agilent, Santa Clara, CA, USA) to assess DNA quality, and 2 µL onto the Qubit^TM^ dsDNA BR Assay Kit (ThermoFisher Scientific, Waltham, MA, USA) to provide an estimate of DNA quantity.

The recovered material from the Illumina lysis kit (30 µL) was fragmented and tagged with NGS adapter sequences by incubating on a thermal cycler with 10 µL PCR-Free Bead-Linked Transposomes (BLT-PF) and 10 µL Tagmentation Buffer 1 (TB1) at 41 °C for 5 min. The reaction was stopped by adding 10 µL of Stop Tagment Buffer 2 (ST2) and incubating for 2 min. Beads were washed with 150 µL of Tagment wash buffer (TWB). Indices were added to adapter sequences using 45 µL of Extension Ligation Mix (ELM) and 5 µL of the appropriate IDT index adapter, then incubating them for 5 min each at 37 °C and 50 °C on a thermal cycler. The beads were, again, washed twice, this time with 75 µL of TWB, before elution from the beads in 45 µL diluted HP3. Libraries were cleaned up and double size selected using sequential incubations (36 µL then 42 µL) with Illumina Purification Beads (plus appropriate 80% EtOH washes). The final libraries were eluted in 22 µL of RSB.

Libraries from the extracted DNA were prepared using the same protocol, with a modification to allow for DNA input (350 ng DNA in 25 µL was added to 15 µL BLT-PF and 10 µL Tagmentation Buffer 1; TB1).

### 2.4. Sequencing Library Quantification, Pooling, and Sequencing

All libraries were quantified using Kapa qPCR (Merck Life Science UK, Gillingham, UK) on a Roche LightCycler LC480 (Roche Diagnostics Limited, Burgess Hill, UK) and normalized to 0.65 nM. Normalized libraries were pooled at equal volumes and loaded onto S4 flowcells using Illumina^®^ NovaSeq 6000 Sequencing System Guide XP Workflow. S4 flowcells were sequenced on Novaseq 6000 to generate paired reads of 150 bp.

NovaSeq Control Software version 1.7.5 automatically transferred the base call (*.bcl) files into the BaseSpace Sequencing Hub.

### 2.5. Data Analysis

Data analysis took place on the BaseSpace Sequencing Hub (BSSH). Data were streamed directly to BSSH from the sequencing instrument. Based on previous experience, we predicted that 130 Gbp of sequencing data would typically be required to achieve an aligned and de-duplicated coverage of 40×. Reads were aligned to Human Reference Genome GRCh38 (alt-aware, with HLAs) using the DragenDNA 3.9.5 analysis workflow. Metrics from the DragenDNA workflow were plotted using R and Python.

Variants which were likely pathogenic were identified by filtering variants called by the DragenDNA workflow using TruSight^TM^ Software Suite (Illumina Inc., San Diego, CA, USA), software that allows rapid analysis and interpretation of the discovered variants. Each experimental condition from each participant was imported as an individual case in the software. Filters were applied in the Interpretation tab of the software to show only PASS variants in or around the gene of interest (i.e., variants occurring in or around *CFTR* for cystic fibrosis and *PAH* for phenylketonuria). We initially searched for two heterozygous variants or one homozygous variant described as pathogenic or likely pathogenic in ClinVar and confirmed the variants discovered by visual inspection by reviewing the aligned reads (bam file) in IGV. For one of the PKU patients, this strategy only revealed a single heterozygous variant. We extended the search to intronic regions of *PAH* where the software had identified a Translocation Breakend variant. Manual review of this variant in IGV confirmed the presence of reads consistent with the presence of a 4.2 Mb inversion.

## 3. Results

### 3.1. High-Quality WGS Data Are Achieved from DBS

Library preparation protocols for DNA sequencing can proceed directly from DBS with the inclusion of a simple lysis step, negating the requirement for DNA extraction from blood. DNA sequencing libraries were made initially from both DBS punches (6 × 3 mm) and extracted DNA (350 ng) using the Illumina DNA PCR-Free library preparation kit following the manufacturer’s standard protocol (Appendix A). PCR-Free WGS libraries were successfully generated from both sample inputs for all individuals (n = 20) and combined in pools for sequencing. We assessed the relative quality of the genomes obtained by several metrics generated using the DRAGEN (Dynamic Read Analysis for GENomics) analysis pipeline (Illumina Inc., San Diego, CA, USA). The achieved coverage was proportional to the yield of sequencing rather than the input material (Median 44.7×, range 29.0×–66.3×; Appendix A). At this level of sequencing coverage, on average, there are enough data to re-sequence each autosomal position more than 40 times; however, the actual depth of coverage at each position varies due to the localized sequence content, with some duplicated regions of the genome causing problems with alignment [36,37,38]. Sampling a position more than 20 times gives us an excellent chance of seeing any heterozygous variant present at that position. More than 95% of the autosomes and >97.5% of the coding exons were covered by 20 reads or more in the majority of samples (Figure 1a,b, Appendix A). Irregularities or biases in the expected sequence representation (as described by sequence coverage variability in regions with different proportions of adenine and thymine bases versus guanine and cytosine bases; GC bias) can be used to identify poor sample or sequencing quality. GC bias curves were plotted for DNA and DBS inputs and show the expected normalized coverage (i.e., a value close to ‘1’) across the majority of the genome when grouped by sequence content (Figure 1c), demonstrating that high-quality WGS data were generated regardless of input.

### 3.2. Good-Quality WGS Data Can Be Achieved from a Variety of DBS Punch Configurations

The material collected on DBS cards is normally used as an input for a number of assays. Given that this precious resource can be in short supply, we wanted to investigate whether there was an impact of reducing the number of punches used as input to the WGS assay from the recommended 6 × 3 mm punches (see Methods). Remaining mindful that some laboratories use a 6 mm punch head for processing DBS, we also wanted to explore how well sub-punches of this size performed in the assay. We returned to the DBS cards taken from participants 9 to 20 to test additional punch configurations (3 × 3 mm, 1 × 3 mm, and 1 × 6 mm) as inputs for comparison.

Altering the punch configuration could result in a change in the amount of input material into WGS library preparation, as there is considerable variation in the volume of blood (9 μL, 3 μL, and 12 μL, respectively [39,40]), and, therefore, the DNA available in each configuration. While all modifications led to some reduction in input material and sequencing library yield, there was sufficient material to generate PCR-Free libraries in each case (Figure 2a). Once again, the coverage which was achieved was proportional to the sequencing yield, with no difference in sequencing quality and only subtle differences in normalized GC content (Figure 2b–d).

While high-quality WGS data were achieved from all input types, we were able to detect subtle differences across the gradient of input types, which might determine the appropriate sub-punch configuration. The fragment length reduced as fewer 3 mm punches were added to the library preparation, consistent with lower DNA input and resulting in shorter library insert sizes for these samples (Figure 2a,e). This is not unexpected given the increased ratio of transposomes to DNA molecules, which would result in a higher degree of fragmentation of the DNA. Insert size (or library fragment length) is an important consideration because sequencing reads are generated from both ends of each fragment of the library in standard (paired-end) Illumina sequencing. Thus, libraries with fragment lengths that are too short can lead to reads with significant overlap, resulting in the generation of redundant information. Short fragment lengths can also result in read pairs that are more challenging to align to the human reference genome sequence, especially those within short, interspersed repeat elements and other sequences that display significant similarity to others present in the genome. While the reduction in library fragment length seen here is not large enough to cause significant impact, more sequencing was required in order to achieve a minimum depth of 20x than for other punch configurations (Figure 2b,c). Library diversity, i.e., the number of unique fragments contained in the library, may therefore be the main driver behind the need to perform additional sequencing of some of the lower-input libraries to achieve the desired level of coverage. Interestingly, the 1 × 6 mm sub-punch inputs yielded libraries of similar fragment sizes to standard 6 × 3 mm despite the lower input, as estimated by the Qubit assay. This may suggest that the Qubit estimation of DNA content in the crude lysate extraction is not as accurate as that of traditionally extracted DNA, but it is interesting to note that the library yield and estimated volume of blood added (via the DBS) were also closer for the 3 × 3 mm and 1 × 6 mm sub-punch configurations.

### 3.3. Identification of Clincally Relevant Variants from DBSs

Three of the participants included in the study (6, 7, and 8) have cystic fibrosis. WGS data were generated from the extracted DNA, and 6 × 3 mm DBS sub-punch inputs from each of these participants and variants were filtered using TruSight^TM^ Software Suite (TSS) to search for high-confidence, likely pathogenic variants. We were able to identify two such variants for each participant. Participant 6 was homozygous for loss of the phenylalanine codon at position 508 of the *CFTR* gene (c.1521_1523del ΔF508), the variant most commonly associated with cystic fibrosis. Participants 7 and 8 were heterozygous for this deletion, but carried additional missense variants (c.3909 C > G Asn1303Lys and c.1624 G > T Gly542Ter, respectively). Likely pathogenic variants in *CFTR* identified by the variant caller were reviewed and confirmed using the Integrative Genomics Viewer (IGV [41,42]; Figure 3). These data demonstrate that clinically relevant variants can be identified from DBS and are in concordance to those variants detected in the peripheral blood.

### 3.4. Detection of Clinically Relevant Variants from DBS Taken Directly from the Participant

To facilitate direct comparison, all previous DBS were formed using the same EDTA blood specimen that was used for DNA extraction from peripheral blood. To confirm that the blood applied directly to a collection card by capillary blood collection (using a finger prick) would show an equivalent performance, we tested samples from two individuals with phenylketonuria (PKU). DBS were collected by capillary taken directly from the participant as part of routine monitoring, as well as being generated from an EDTA blood draw. The metrics obtained from the libraries generated using both inputs were highly comparable (Figure 4a–c).

Variants identified from these participants were filtered using TSS in order to identify variants of potential clinical significance. Two variants of interest were identified in each participant regardless of assay input (DBS from patient, DBS via EDTA blood draw, and extracted DNA). One participant carried a homozygous deletion in *PAH*, c.558_559del, which is predicted to result in protein truncation, while the other carried both a heterozygous missense variant in PAF, c.204A > T, and a heterozygous 4.2 Mb inversion displacing exons 8–14 of *PAH* (Figure 4d,e). The identification of the latter demonstrates the power of WGS (from either input) to detect clinically relevant complex structural variants.

## 4. Discussion

The use of PCR-Free library preparation for WGS is desirable, because PCR amplification steps can introduce coverage biases across the genome and even introduce incorrect bases during amplification [43,44,45]. In this study, we demonstrate that PCR-Free WGS can be successfully performed directly from a range of DBS inputs (6 × 3 mm, 3 × 3 mm, 1 × 3 mm, 1 × 6 mm) extending beyond those recommended by the manufacturer (6 × 3 mm). The obtained data were highly comparable to those generated using extracted DNA from peripheral blood and were not dramatically affected by the size or number of the sub-punches analyzed. The ability to perform a PCR-Free free library preparation for all sub-punch configurations demonstrates that the ‘gold standard’ WGS library preparation method is accessible across a range of DBS inputs.

A number of metrics can be used to assess WGS data quality. Here, we focused on depth of coverage because, ultimately, this has the most significant impact on the ability to identify variants in mature NGS pipelines. Our study adds to the growing body of evidence that DBS are a valid input for NGS, offering a real alternative to DNA extracted from blood, where sample collection is challenging [29,30,31,32,33]. Previous studies have investigated the quantity and quality of DNA from DBS before library preparation. The protocol used here is designed to proceed directly from DBS, releasing and lysing cells from the collection card using a simple procedure before starting NGS library construction. While the protocol does not require an assessment of DNA quality, and indeed, QC values may not be comparable to those of DNA extracted by a standard DNA extraction kit, assessment of the material going into the NGS library preparation by gDNA TapeStation suggests that the DNA going into the assay is, in fact, of high quality (Appendix A).

While there was no significant impact on data quality when reducing the number of sub-punches, laboratories seeking to do so should be aware that performing optional steps in the protocol may be required in order to achieve a successful WGS workflow. For example, the Illumina DNA library prep protocol can remove the requirement for quantitative PCR of libraries in advance of loading on the sequencing instrument, because the bead linked transposomes are able to normalize the amount of library produced in each reaction. This is desirable when pooling libraries from multiple samples to be run together at the same time on the sequencing instrument to ensure that equal amounts of data are generated from each sample. However, the reduced DNA input from lower numbers of sub-punches may mean that the library normalization properties of the bead-linked transposomes will be lost; thus, assessment of library quantity by quantitative PCR should be maintained to ensure accurate pooling and evenness of coverage. If a single 3 mm punch is used, then additional sequencing may be required in order to achieve the desired level of coverage, owing to the smaller library fragment sizes. Therefore, the gain of a streamlined DNA punching pipeline may have some impact on cost and turnaround time in the genomic laboratory. The obtained data suggest that a 1 × 6 mm sub-punch may provide the best combination of simplicity and WGS quality.

The ability to generate PCR-Free whole genomes from DBS is particularly exciting considering that DBS are already collected routinely as part of routine care in newborn screening programs across the world. Material from these DBS is already used as input in a variety of assays, which means that availability of materials could be limited for further testing. The fact that this workflow can proceed from just a single blood sub-punch opens the possibility of performing WGS in parallel to existing tests, adding extra detail to information captured by the current programs or accelerating the identification of genetic changes that would normally require a secondary sample collection procedure (as is the situation today with cystic fibrosis testing). Another challenge of adopting WGS for NBS is the interpretation of the revealed variants. Many of these have unknown significance, especially when studying diverse populations where there is a need for improved reference databases “https://www.ga4gh.org/” (accessed on 31 August 2023), “https://h3africa.org/” (accessed on 31 August 2023) [46,47,48,49,50]. Additionally, recent work using primate sequenced and machine learning may provide valuable information as to whether variants of unknown significance are likely to be pathogenic or benign [51,52]. The individuals in this study had known conditions, allowing us to focus our search for likely pathogenic variants in a limited number of well-described disease genes.

While other NGS assays could proceed from DBS, we believe that the whole-genome sequencing test is the appropriate one in this scenario. Firstly, the non-targeted nature of the WGS assay provides maximum ‘future proofing’, as clinically relevant genes or regions can be added to reports as they are discovered, without modification to the sample collection or laboratory assay workflows. Secondly, a WGS library is ready for sequencing in a significantly shorter amount of time than one obtained using enrichment strategies, which require additional steps after library preparation. And thirdly, the comprehensive nature of the WGS test means that all classes of variant can be detected, including complex structural variants like the *PAH* inversion discovered in this study.

A WGS test would be a powerful addition to any newborn screening program, potentially providing information on the genetic health of the individual during their lifetime. There are significant challenges in the logistics of sample collection and processing, and the ability to utilize existing testing infrastructures with a proven track record of success is incredibly attractive.

## Figures and Tables

**Figure 1 IJNS-09-00052-f001:**
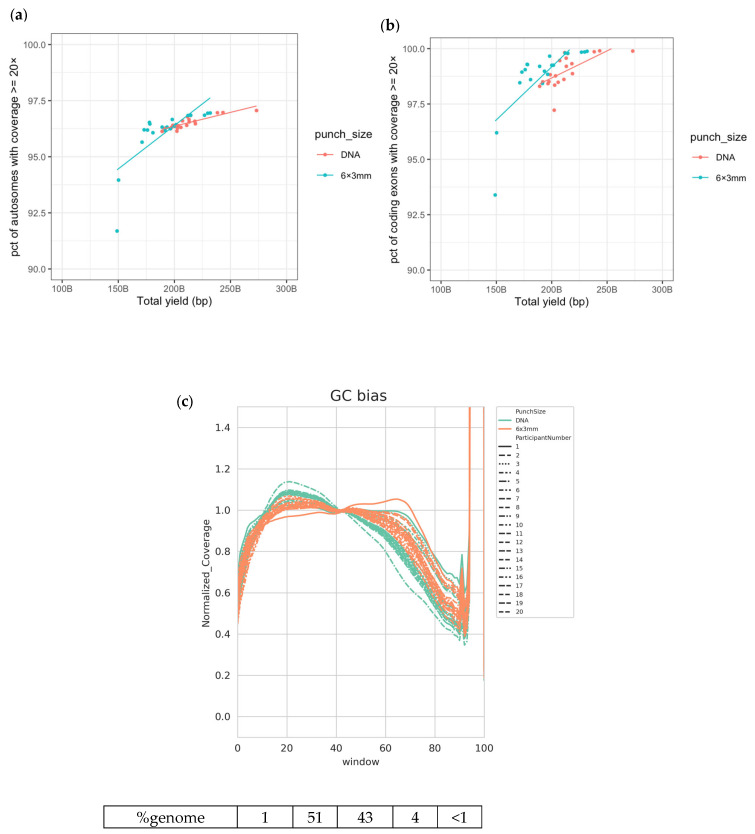
Good-quality WGS data achieved from DBS and extracted DNA. (**a**) Percentage of autosomes with coverage ≥20× vs. total yield sequenced; (**b**) percentage of coding exons with coverage ≥20× vs. total yield sequenced; (**c**) normalized coverage across various windows of GC content in the human genome, grouped by participant. The numbers below the plot indicate the percentage of the human reference genome sequence included in each window of GC content.

**Figure 2 IJNS-09-00052-f002:**
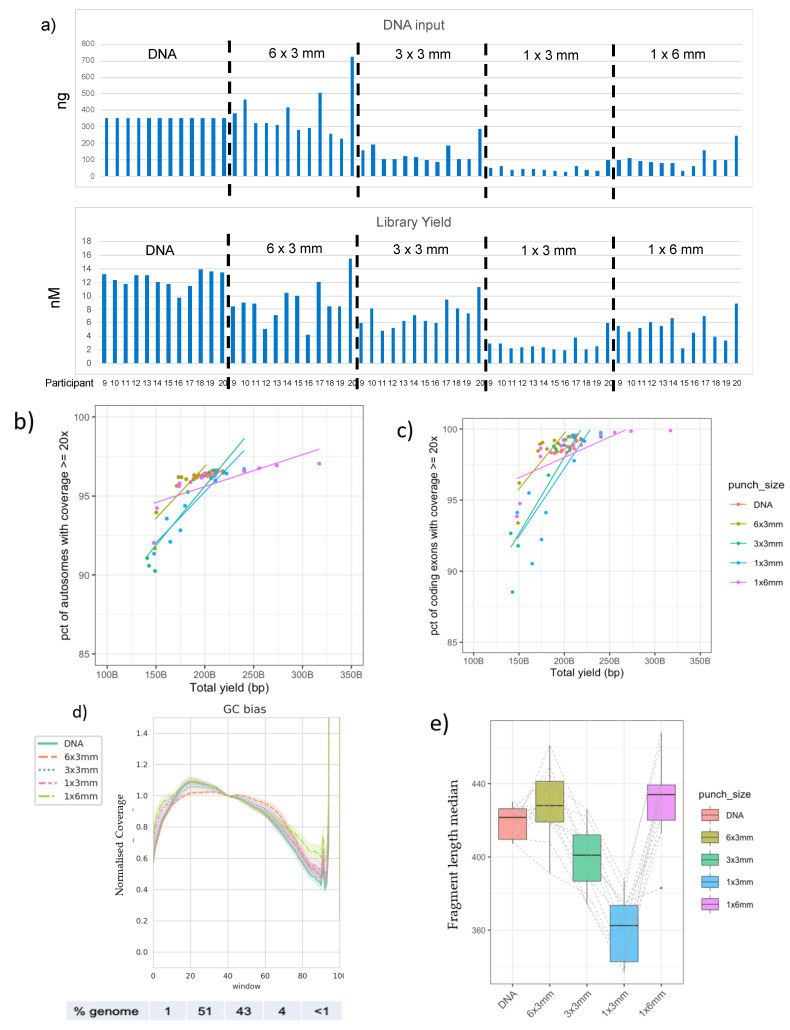
WGS data generated from a variety of DBS punch inputs. (**a**) Qubit values measured post-DBS lysis protocol for participants 9 to 20 are shown above the sequencing library quantifications determined by quantitative PCR (qPCR) for each input type. (**b**) Percentage of autosomes with coverage ≥20× vs. total yield sequenced; (**c**) percentage of coding exons with coverage ≥20× vs. total yield sequenced; (**d**) normalized coverage across various windows of GC content in the human genome, grouped by sample input type and averaged for the 20 participants. Numbers below the plot indicate the percentages of genomic sequences included in each window of GC content; (**e**) fragment length median for each input type for participants 9 to 20. Ranges of values are summarized by box and whisker plots, with dotted lines link data points from the same participants.

**Figure 3 IJNS-09-00052-f003:**
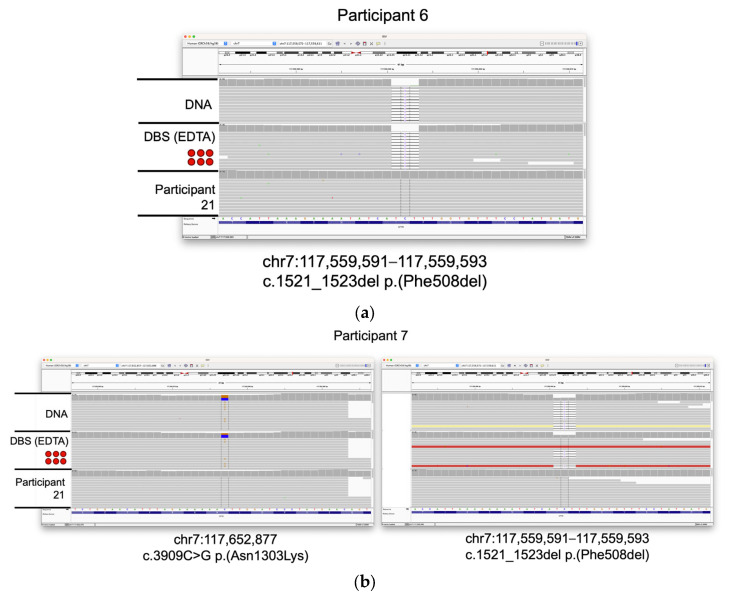
Identification of *CFTR* variants in samples from individuals with cystic fibrosis. IGV screenshots of likely pathogenic variants discovered in three participants with cystic fibrosis. Tracks from DNA and DBS (6 × 3 mm) inputs are shown. Data from participant 21 (extracted DNA) are shown for comparison. (**a**) Homozygous Phe508del variant seen in participant 6. (**b**) Asn1303Lys and Phe508del variants seen in participant 7. (**c**) Gly542Ter and Phe508del variants seen in participant 8.

**Figure 4 IJNS-09-00052-f004:**
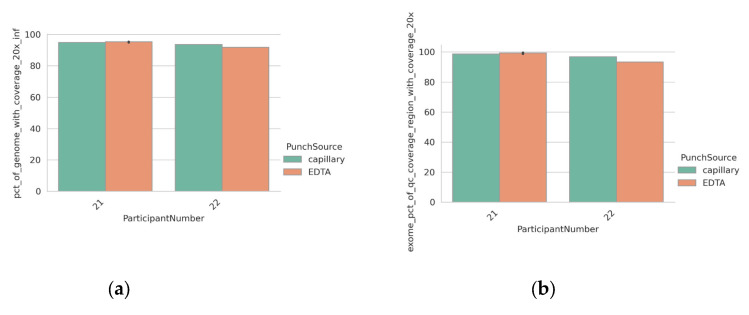
Equivalent performance of a DBS punch (6 mm) and DNA extracted via EDTA blood tubes. (**a**) Percentage of autosomal regions with ≥20× coverage for each sample; (**b**) percentage of exome regions with ≥20× coverage for each sample; (**c**) normalized coverage across various windows of GC content in the human genome, grouped by sample input type; (**d**) IGV screenshot of likely pathogenic gene variants discovered in participant with PKU. Tracks from the DNA and DBS inputs are shown. Data from participant 6 (extracted DNA) are shown for comparison (**e**) Likely pathogenic variants discovered in samples taken from a second participant with PKU. Cartoon representation of the variant location on chromosome 12 and zoomed-in IGV screenshots from the DBS taken directly from the patient. Asterix highlights the presence of the SNV shown in lower panel.

## Data Availability

The data presented in this study are available on request from the corresponding author.

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
