# Peer review of "Whole-Genome Sequencing Can Identify Clinically Relevant Variants from a Single Sub-Punch of a Dried Blood Spot Specimen"

_2409-515X, 2023, doi:10.3390/ijns9030052_

Round 1

Reviewer 1 Report

Wonderful article with robust data and excellent graphics. Some suggestions:

Additional References - Section Introduction:

Line 52: Consider adding more references about the impact of the diagnostic odyssey in rare diseases including:  (1) https://everylifefoundation.org/burden-study/; (2) https://bmchealthservres.biomedcentral.com/articles/10.1186/s12913-023-09886-7; (3) https://pubmed.ncbi.nlm.nih.gov/36331261/. 

Line 56 and 66: Consider adding more references about the use of WGS in neonatal sequencing including: (1) https://www.ncbi.nlm.nih.gov/pmc/articles/PMC5260149/; (2) https://pubmed.ncbi.nlm.nih.gov/32853555/.

Additional Content: (1) Line 46 to 48 - Consider describing and citing limitations of WGS to detect CNV and SVs; (2) Line 48 to 50  - Consider further explanation of rapid expansion of information noting that not only does laboratory methods of WGS from DBS need to improve and be validated, so does the interpretation and data analysis approaches; (3) Line 509 or elsewhere in the discussion, note the lack of reference databases of sequence for diverse populations and the case study of CF and PKU limit the exploration of novel variants of unknown significance.  

Author Response

Thank you very much for taking the time to review this manuscript, we are delighted you enjoyed it.

Responses to comments are below, I've also attached the updated manuscript with changes highlighted:

Additional References:

Line 52: Consider adding more references about the impact of the diagnostic odyssey in rare diseases including:  (1) https://everylifefoundation.org/burden-study/; (2) https://bmchealthservres.biomedcentral.com/articles/10.1186/s12913-023-09886-7; (3) https://pubmed.ncbi.nlm.nih.gov/36331261/.

We are very happy to include the additional references (1) and (2) and have added these to the text along with some minor modifications to the text to improve readability around their inclusion. We felt reference (3) has less of a focus on impact to diagnostic odyssey, and hope the following additions are appropriate instead:

  1. 100,000 Genomes Pilot on Rare-Disease Diagnosis in Health Care — Preliminary Report. New England Journal of Medicine, 2021. 385(20): p. 1868-1880.
  2. Saunders, C.J., et al., Rapid whole-genome sequencing for genetic disease diagnosis in neonatal intensive care units. Sci Transl Med, 2012. 4(154): p. 154ra135.
  3. Group, N.I.S., et al., Effect of Whole-Genome Sequencing on the Clinical Management of Acutely Ill Infants With Suspected Genetic Disease: A Randomized Clinical Trial. JAMA Pediatr, 2021. 175(12): p. 1218-1226.
  4. Clark, M.M., et al., Diagnosis of genetic diseases in seriously ill children by rapid whole-genome sequencing and automated phenotyping and interpretation. Sci Transl Med, 2019. 11(489).
  5. Kingsmore, S.F., et al., A Randomized, Controlled Trial of the Analytic and Diagnostic Performance of Singleton and Trio, Rapid Genome and Exome Sequencing in Ill Infants. Am J Hum Genet, 2019. 105(4): p. 719-733.
  6. Stranneheim, H., et al., Integration of whole genome sequencing into a healthcare setting: high diagnostic rates across multiple clinical entities in 3219 rare disease patients. Genome Medicine, 2021. 13(1): p. 40.
  7. Bowdin, S.C., et al., The SickKids Genome Clinic: developing and evaluating a pediatric model for individualized genomic medicine. Clin Genet, 2016. 89(1): p. 10-9.

Line 56 and 66: Consider adding more references about the use of WGS in neonatal sequencing including: (1) https://www.ncbi.nlm.nih.gov/pmc/articles/PMC5260149/; (2) https://pubmed.ncbi.nlm.nih.gov/32853555/.

We are very happy to include the additional references (1). As reference (2) demonstrates whole exome sequencing in newborns and our manuscript is on WGS we wondered if it might be more appropriate to show additional references from WGS studies and so now include the below.

  1. Group, N.I.S., et al., Effect of Whole-Genome Sequencing on the Clinical Management of Acutely Ill Infants With Suspected Genetic Disease: A Randomized Clinical Trial. JAMA Pediatr, 2021. 175(12): p. 1218-1226.
  2. Petrikin, J.E., et al., The NSIGHT1-randomized controlled trial: rapid whole-genome sequencing for accelerated etiologic diagnosis in critically ill infants. NPJ Genom Med, 2018. 3: p. 6.
  3. Kingsmore, S.F. and T.B. Consortium, Dispatches from Biotech beginning BeginNGS: Rapid newborn genome sequencing to end the diagnostic and therapeutic odyssey. American Journal of Medical Genetics Part C: Seminars in Medical Genetics, 2022. 190(2): p. 243-256.
  4. Kingsmore, S.F., et al., A genome sequencing system for universal newborn screening, diagnosis, and precision medicine for severe genetic diseases. The American Journal of Human Genetics, 2022. 109(9): p. 1605-1619.
  5. Berg, J.S., et al., Newborn Sequencing in Genomic Medicine and Public Health. Pediatrics, 2017. 139(2).
  6. Seydel, C., Baby’s first genome. Nature Biotechnology, 2022. 40(5): p. 636-640.
  7. Dimmock, D., et al., Project Baby Bear: Rapid precision care incorporating rWGS in 5 California children's hospitals demonstrates improved clinical outcomes and reduced costs of care. Am J Hum Genet, 2021. 108(7): p. 1231-1238.
  8. Mestek-Boukhibar, L., et al., Rapid Paediatric Sequencing (RaPS): comprehensive real-life workflow for rapid diagnosis of critically ill children. Journal of Medical Genetics, 2018. 55(11): p. 721-728.
  9. Ding, Y., et al., Scalable, high quality, whole genome sequencing from archived, newborn, dried blood spots. npj Genomic Medicine, 2023. 8(1): p. 5.
  10. Agrawal, P., et al., Validation of whole genome sequencing from dried blood spots. BMC Medical Genomics, 2021. 14(1): p. 110.
  11. Poulsen, J.B., et al., High-Quality Exome Sequencing of Whole-Genome Amplified Neonatal Dried Blood Spot DNA. PLoS One, 2016. 11(4): p. e0153253.
  12. Hollegaard, M.V., et al., Archived neonatal dried blood spot samples can be used for accurate whole genome and exome-targeted next-generation sequencing. Molecular Genetics and Metabolism, 2013. 110(1): p. 65-72.
  13. Bassaganyas, L., et al., Whole exome and whole genome sequencing with dried blood spot DNA without whole genome amplification. Human Mutation, 2018. 39(1): p. 167-171.
  14. Roman, T.S., et al., Genomic Sequencing for Newborn Screening: Results of the NC NEXUS Project. Am J Hum Genet, 2020. 107(4): p. 596-611.

Additional Content:

(1) Line 46 to 48 - Consider describing and citing limitations of WGS to detect CNV and SVs;

Text modified below:

WGS is the most comprehensive test available to capture all classes of variants, from small events such as single nucleotide variants (SNVs) and small indels, to larger changes in genetic material such as copy number changes (CNV) [5, 6] and structural variants (SVs) [7]; the last two often originate deep in intronic regions not included on gene panels or whole-exome sequencing (WES). Calling SVs, even from WGS data, can be compromised by the presence of repetitive or simple sequence involved in the variant, but there has been success in addressing such cases using combinations of callers or custom callers targeted at specific genomic regions [8-10].

  1. Gross, A.M., et al., Copy-number variants in clinical genome sequencing: deployment and interpretation for rare and undiagnosed disease. Genet Med, 2019. 21(5): p. 1121-1130.
  2. Whitford, W., et al., Evaluation of the performance of copy number variant prediction tools for the detection of deletions from whole genome sequencing data. Journal of Biomedical Informatics, 2019. 94: p. 103174.
  3. Austin-Tse, C.A., et al., Best practices for the interpretation and reporting of clinical whole genome sequencing. npj Genomic Medicine, 2022. 7(1): p. 27.
  4. Liu, Z., et al., Towards accurate and reliable resolution of structural variants for clinical diagnosis. Genome Biology, 2022. 23(1): p. 68.
  5. Coutelier, M., et al., Combining callers improves the detection of copy number variants from whole-genome sequencing. European Journal of Human Genetics, 2022. 30(2): p. 178-186.
  6. Dolzhenko, E., et al., ExpansionHunter Denovo: a computational method for locating known and novel repeat expansions in short-read sequencing data. Genome Biology, 2020. 21(1): p. 102.

(2) Line 48 to 50  - Consider further explanation of rapid expansion of information noting that not only does laboratory methods of WGS from DBS need to improve and be validated, so does the interpretation and data analysis approaches;

Text modified below:

The algorithms used to capture variants and distil the data into a clinical format are under continual improvement, with updates to analysis pipelines requiring validation as well as the laboratory workflows. While this rapid expansion of information can seem daunting, the content is incredibly powerful for aiding diagnosis Indeed, in rare genetic disorders, whole-genome sequencing (WGS) has been established in numerous healthcare systems as a primary test, in a bid to lessen the diagnostic odyssey faced by patients with conditions challenging to diagnose [4, 11-16].

(3) Line 509 or elsewhere in the discussion, note the lack of reference databases of sequence for diverse populations and the case study of CF and PKU limit the exploration of novel variants of unknown significance. 

Text modified below:

The ability to generate PCR-Free whole genomes from DBS is particularly exciting considering that DBS are already collected routinely as part of routine care in newborn screening programmes across the world. Material from these DBS is already used as input in a variety of assays, which means that availability of material could be limited for performing further testing . The fact that this workflow can proceed from just a single blood sub-punch opens the possibility of performing WGS in parallel to existing tests, adding extra detail to information captured by current programmes, or accelerating the identification of genetic changes that would normally require a secondary sample collection procedure (as is the situation today with Cystic Fibrosis testing). Another challenge to adopting WGS for NBS is the interpretation of the variants revealed. Many of these have unknown significance, especially when studying diverse populations where there is a need for improved reference databases (https://www.ga4gh.org/, https://h3africa.org/, [46-50]). Additionally, recent work using primate sequenced and machine learning may provide valuable information as to whether variants of unknown significance are likely to be pathogenic or benign [51, 52]. The individuals in this study had a known condition, allowing us to focus our search for likely pathogenic variants in a limited number of well described disease genes.

  1. Mallick, S., et al., The Simons Genome Diversity Project: 300 genomes from 142 diverse populations. Nature, 2016. 538(7624): p. 201-206.
  2. Abul-Husn, N.S., et al., Implementing genomic screening in diverse populations. Genome Medicine, 2021. 13(1): p. 17.
  3. Abul-Husn, N.S., et al., Exome sequencing reveals a high prevalence of BRCA1 and BRCA2 founder variants in a diverse population-based biobank. Genome Medicine, 2019. 12(1): p. 2.
  4. Brnich, S.E., et al., Recommendations for application of the functional evidence PS3/BS3 criterion using the ACMG/AMP sequence variant interpretation framework. Genome Medicine, 2019. 12(1): p. 3.
  5. Rehm, H.L. and D.M. Fowler, Keeping up with the genomes: scaling genomic variant interpretation. Genome Medicine, 2019. 12(1): p. 5.
  6. Gao, H., et al., The landscape of tolerated genetic variation in humans and primates. Science, 2023. 380(6648): p. eabn8153.
  7. Fiziev, P.P., et al., Rare penetrant mutations confer severe risk of common diseases. Science, 2023. 380(6648): p. eabo1131.

Reviewer 2 Report

The paper describes identification of clinically relevant variants from a single sub-punch of dried blood spot specimen using whole-genome sequencing.  The authors compared WGS metrics obtained from libraries generated directly from DBS to those generated from DNA extracted from peripheral blood. They explored the flexibility of DBSS as input to WGS by altering punch number and size as input to the assay. They showed that WGS libraries can be successfully generated from a variety of DBS inputs with equivalent data quality observed across a number of key metrics of importance in the detection of gene variants. They did not observe differences in the performance of DBS versus peripheral blood  extracted DNA in the detection of likely pathogenic variants in cystic fibrosis and PKU individuals. They conclude that WGS can be performed directly from DBS and is a powerful method for rapid discovery of clinically relevant disease-causing gene variants.

The paper is well written and the methodology as described will lend itself to timely confirmation of newborns who have actionable inborn errors of metabolism without resorting blood draws that can be challenging at that age.

Author Response

Thank you very much for taking the time to review this manuscript, we are delighted you enjoyed it!

Author Response

Thank you very much for taking the time to review this manuscript, please find our responses to your comments below. I've also attached an updated version of the manuscript with changes highlighted

  1. Materials and Methods

Line95: Please provide a citation or a brief explanation of the KUVAN responsiveness testing pathway.

Thank you for spotting this, the reference has been added to the manuscript.

Burton, B.K., et al., The response of patients with phenylketonuria and elevated serum phenylalanine to treatment with oral sapropterin dihydrochloride (6R-tetrahydrobiopterin): a phase II, multicentre, open-label, screening study. J Inherit Metab Dis, 2007. 30(5): p. 700-7.

Also, please describe storage conditions and storage time until analysis for the collected samples.

Thank you for spotting this, last sentence of section updated to include these conditions 

To compare WGS data generated from liquid whole and DBS specimens we retrieved anonymized residual blood specimens collected into EDTA blood tubes. An aliquot of the liquid blood specimen was taken and 50 ml aliquots applied to filter paper collection devices (PerkinElmer 226). Liquid whole blood specimens from the PKU patients were collected during routine monitoring and genetic analysis as part of the KUVAN responsiveness testing pathway [35], an additional sample was collected from these patients by spotting blood from a finger prick directly on to the blood collection card. Blood spot cards were stored at room temperature for 1 to 6 weeks before processing.

  1. Results

Line177 “There was little difference...” Please briefly summarize the differences in quality between 6x3mm DBS input and DNA from whole blood.

We have removed this sentence from the manuscript to improve clarity. The following passage of text describes the data.

Line244 Please provide a reference for the recommendation of using 6x3mm DBS punches.

We felt that this information was more appropriate for the material and methods section so have added it there and referred to the methods section on Line 244.

Sequencing libraries were prepared using the Illumina DNA PCR-Free Prep kit (Illumina Inc., San Diego, CA, USA) which proceeds directly from six 3.2 mm DBS sub-punches when following the standard protocol (Illumina DNA PCR-Free Library Prep Reference Guide, Illumina document #1000000086922 v03). The appropriate number of 3.2 mm or 6 mm sub-punches (6, 3 or 1) were obtained from each DBS using a DBS Puncher® Instrument (Perkin Elmer, Waltham, MA, USA).

Line251 Please provide citations on the estimation of blood volume in DBS punches.

The following citations have been added to the manuscript

Moat, S.J., R.S. George, and R.S. Carling, Use of Dried Blood Spot Specimens to Monitor Patients with Inherited Metabolic Disorders. Int J Neonatal Screen, 2020. 6(2): p. 26.

Moat, S.J., et al., Development of a high-throughput SARS-CoV-2 antibody testing pathway using dried blood spot specimens. Ann Clin Biochem, 2021. 58(2): p. 123-131.

Information provided in Figures 2b-d could be slightly larger to be more reader-friendly, preferably like Fig 1a-c. The data from different punch sizes are currently difficult to compare.

Apologies, I’ve had a go at this but very happy to work with the journal to improve readability in the final document.

Moreover, identification of rare variants is shown for sequencing of 6x3mm punch sample and extracted DNA sample. The authors should describe if sequencing results from the other sample inputs (3x3 mm, 1x3mm and 1x6mm DBS punches) were able to identify rare variants or if there were limitations for these sample types. Perhaps adding results for these in supplementary data section would be beneficial to the reader.

Unfortunately we did not process the full set of sub-punch configurations for DBS from CF or PKU patients. We were initially interested in comparing the extracted DNA to the standard DBS input (6x3mm) comparison in the CF samples. It was only after this work was complete that we explored altering the sub-punch number and size. After we had determined that a 1x 6mm punch should be sufficient, we performed the extract DNA vs 1x6mm sub-punch experiments on the PKU patient samples. As this was successful we did not process any further material.  

Supplementary Materials Figure S3

Axis text need to be larger as well as header text to be visible, or removed from plots since these are explained in the figure text.

Apologies once again, I’ve had a go at this but very happy to work with the journal to improve readability in the final document.

The comparison is clearly described for preparing gDNA libraries however, it is not described for interpretating WGS data like identification of rare variants from the sequencing of the libraries. This would add important insight to the manuscript.

This is an excellent point. We’ve added some detail to the methods section below in an attempt to address this.

2.5 Data Analysis

            Data analysis took place on BaseSpace Sequencing Hub (BSSH). Data were streamed directly to BSSH from the sequencing instrument. Based on previous experience, we predicted that 130 Gbp of sequencing data would typically be required to achieve an aligned and de-duplicated coverage of 40x. Reads were aligned to the Human hg38 alt-aware with HLAs reference sequence using the DragenDNA 3.9.5 analysis workflow. Metrics from the DragenDNA workflow were plotted using R and python.

            Likely pathogenic variants were identified by filtering variants called by the DragenDNA workflow using TruSightTM Software Suite (Illumina Inc., San Diego, CA, USA), software that allows rapid analysis and interpretation of variants discovered. Each experimental condition from each participant was imported as an individual case in the software. Filters were applied in the Interpretation tab of the software to show only PASS variants in or around the gene of interest (i.e. variants occurring in or around CFTR for Cystic Fibrosis and PAH for Phenylketonuria). We initially searched for two heterozygous variants or one homozygous variant described as pathogenic or likely pathogenic in ClinVar and confirmed the variants discovered by visual inspection by reviewing the aligned reads (bam file) in IGV. For one of the PKU patients this strategy only revealed a single heterozygous variant. We extended the search to intronic regions of PAH where the software had identified a Translocation Breakend variant. Manual review of this variant in IGV confirmed the presence of reads consistent with the presence of a 4.2 Mb inversion.
